# 𝒳-PART: HIGH FIDELITY AND STRUCTURE COHERENT SHAPE DECOMPOSITION

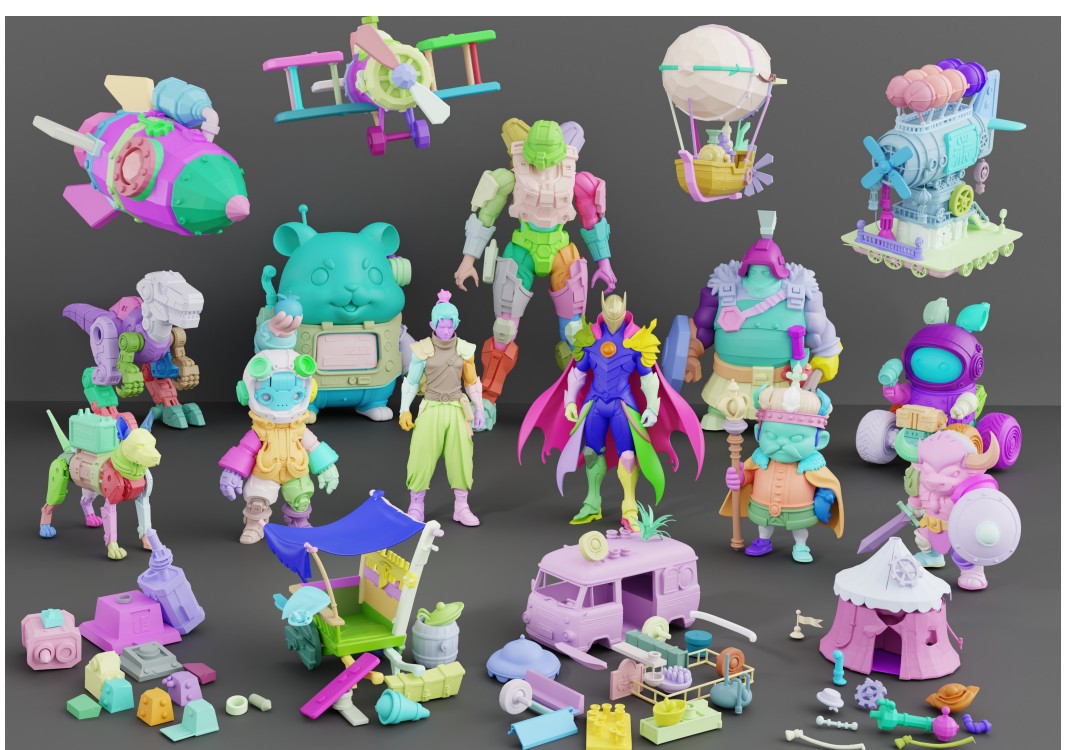

## ABSTRACT

Generating 3D shapes at part level is pivotal for downstream applications such as mesh retopology, UV mapping, and 3D printing. However, existing part-based generation methods often lack sufficient controllability and produce semantically inconsistent decompositions. To this end, we introduce 𝒳-Part, a diffusion-based method designed to decompose a holistic 3D object into semantically meaningful and structurally coherent parts with high geometric fidelity. 𝒳-Part exploits bounding boxes as prompts for part generation and injects point-wise semantic features for meaningful decomposition. Furthermore, we design a pipeline for interactive part editing. Extensive experimental results show that 𝒳-Part significantly advances the state-of-the-art in both part shape quality and semantic correctness. This work establishes a new paradigm for creating production-ready, editable, and structurally sound 3D assets. Codes will be released for public research.

## 1  INTRODUCTION

3D assets are now extensively utilized across a wide range of fields, including gaming, film production, 3D printing, autonomous driving, and robotic simulation. However, traditional 3D content

creation remains a time-consuming process that demands significant expertise. Recent advances in generative AI have substantially lowered the barriers to 3D content generation, particularly with the emergence of foundational 3D models Zhang et al. (2024); Zhao et al. (2025); Lai et al. (2025).

Despite this progress, most existing generative approaches are only capable of producing monolithic 3D models, which poses considerable limitations for practical 3D creation pipelines. Decomposing a complete 3D shape into meaningful semantic parts would greatly facilitate various downstream tasks. For instance, breaking down a complex geometry into simpler parts can significantly ease the process of mesh re-topology Weng et al. (2025) and uv-unwrapping Li et al. (2025a). Generating shapes at the part level presents two major challenges: 1) The decomposed geometry must maintain meaningful part-level semantics, and 2) The generation process must recover geometrically plausible structures for internal regions.

Mainstream part-generation methods adopt the latent vecset diffusion framework Zhang et al. (2023), where each part is represented as an independent set of latent codes for diffusion. The generation process can be executed independently for individual parts (e.g., HoloPart Yang et al. (2025a)) or simultaneously for all parts (e.g., PartCrafter Lin et al. (2025), PartPacker Tang et al. (2025)) with enhanced part synchronization. Furthermore, 2D image segmentation or 3D mesh segmentation are frequently employed for better part generation Chen et al. (2024); Yang et al. (2025a;b). However, these approaches are highly sensitive to inaccuracies in the segmentation results. Alternative works Lin et al. (2025); Tang et al. (2025) do not explicitly rely on segmentation, but they lack controllability and often generate parts with ambiguous boundary.

Motivated by these observations, we present $\mathcal{X}$-Part, a diffusion-based framework that decomposes a holistic mesh into semantically meaningful and structurally coherent 3D parts. The method utilizes the state-of-the-art segmenter $P^3$-SAM Ma et al. (2025) to automatically generate initial part segmentations, bounding boxes, and semantic features. Then the shape decomposition is executed within a synchronized multi-part diffusion process.

Specifically, 1) First, to control part decomposition, instead of directly using segmentation results as input we uses bounding boxes as prompts to indicate part locations and scales. Compared with fine-grained and point-level segmentation cues, bounding boxes provide a coarser form of guidance, which mitigates overfitting to the input segmentation masks. Besides, the bounding box provides additional volume scale information for the partially visible part, benefiting generation and controllability. 2) Second, despite inaccuracies in the segmentation results, we notice that the high-dimension point-wise semantic feature is free from the information compression caused by the mask prediction head used in $P^3$-SAM, resulting in more robust semantic representations. Therefore, we introduce the semantic features from $P^3$-SAM into our diffusion process to guide the multi-part diffusion process. This greatly benefits the part decomposition. 3) Third, we integrate $\mathcal{X}$-Part into a bounding box based part editing pipeline following Lugmayr et al. (2023). It supports local editing, such as splitting a part into several parts and adjusting their scales, to facilitate interactive part generation.

To prove the effectiveness of $\mathcal{X}$-Part, we conducted extensive experiments on various benchmarks. Our results show that $\mathcal{X}$-Part achieves state-of-the-art performance in part-level decomposition and generation. In summary, the contributions of our work are as follows:

1. We propose $\mathcal{X}$-Part, a controllable and editable diffusion framework, capable of generating semantically meaningful and structurally coherent 3D parts.

2. We integrate $\mathcal{X}$-Part into an editable part generation pipeline, which supports multiple interactive editing methods.

3. Extensive experiments demonstrate that $\mathcal{X}$-Part achieves state-of-the-art performance in part-level decomposition and generation.

## 2 RELATED WORK

**Part Segmentation.** The most straightforward approach for decomposing a 3D geometry is segmentation. Conventional methods Qi et al. (2017); Zhao et al. (2021) directly predict per-point semantic labels via supervised learning. However, these methods rely heavily on extensive part-level annotations and generalize poorly beyond seen categories. Inspired by the remarkable success of 2D foundation models like SAM Kirillov et al. (2023) and GLIP Li et al. (2022) in open-vocabulary

tasks, several recent approaches Abdelreheem et al. (2023); Liu et al. (2023); Tang et al. (2024); Thai et al. (2024); Umam et al. (2024); Zhong et al. (2024) attempt to lift 2D visual knowledge to 3D domains. Although these methods improve generalization, they often fail to accurately infer parts in occluded or unobserved regions. To mitigate this, PartField Liu et al. (2025) and SAMPart3D Yang et al. (2024) learn open-world 3D feature fields for semantic part decomposition. P3-SAM Ma et al. (2025) proposes a native 3D part segmentation network trained on a large, purely 3D dataset with part annotations, demonstrating impressive part segmentation results.

**Object-level Shape Generation.** The remarkable success of latent diffusion models in 2D image generation has inspired a new wave of methods extending this capability to 3D object generation. Dreamfusion Poole et al. (2022) introduced Score Distillation Sampling (SDS) to distill 2D priors from pre-trained diffusion models for 3D synthesis, though it often suffers from slow optimization and geometrically inconsistent outputs. Subsequent approaches Li et al. (2023); Long et al. (2024); Shi et al. (2023), reformulated 3D generation as a multi-view image synthesis problem. With the release of large-scale 3D datasets such as Objaverse Deitke et al. (2023b) and Objaverse-XL Deitke et al. (2023a), native 3D generative models have become increasingly prevalent. Methods like 3DShape2VecSet Zhang et al. (2023), Michelangelo Zhao et al. (2023), Clay Zhang et al. (2024), and Dora Chen et al. (2025c) encode object point clouds into vector-set tokens using a variational autoencoder (VAE) Kingma & Welling (2013) and model the distribution via a Diffusion Transformer (DiT) Peebles & Xie (2023). In contrast, Trellis Xiang et al. (2025) employs an explicit voxel representation for coarse geometry and further generates both geometry and appearance from the voxel latents.

**Part-level Shape Generation.** PartGen Chen et al. (2025a) decomposes 3D objects by solving a multi-view segmentation task and subsequently completes and reconstructs each part in 3D. PhyCAGE Yan et al. (2024b) adopt physical regularization for non-rigid part decomposition. While recent methods exploit DiT-based generative methods to achieve part-level generation Yang et al. (2025a); Luo et al. (2025); Lin et al. (2025); Tang et al. (2025); Dong et al. (2025); Yang et al. (2025b); Zhang et al. (2025). HoloPart Yang et al. (2025a) completes part geometry from initial 3D segmentation results. In contrast, PartCrafter Lin et al. (2025) and PartPacker Tang et al. (2025) operate without explicit segmentation, instead leveraging multi-instance DiTs to generate parts automatically. PartPacker Tang et al. (2025) further introduces a dual-volume DiT to model complementary spatial volumes for improved efficiency. Frankenstein Yan et al. (2024a) execute similar idea by packing multiple SDFs in a latent triplane space via VAE. However, these approaches often yield parts with limited geometric quality and offer minimal local controllability. CoPart Dong et al. (2025) incorporates an auxiliary 2D image diffusion model to enhance texture and detail using 2D/3D bounding box conditions, though it supports only up to 8 parts and cannot decompose an existing 3D shape. OmniPart Yang et al. (2025b) adopts an explicit representation similar to Trellis and uses bounding box prompts, yet it lacks the ability to complete occluded geometry. BANG Zhang et al. (2025) frames part generation as an object explosion process, enabling bounding-box-guided decomposition and recursive refinement, but it often fails to preserve fine geometric details throughout the process. AutoPartGen Chen et al. (2025b) employs a latent diffusion model to autoregressively generate parts, which is computationally expensive and offers limited user control.

## 3 METHOD

Our objective is to generate high-fidelity and structure-coherent part geometries from a given object point cloud, while ensuring flexible controllability over the decomposition process. To this end, we propose $\mathcal{X}$-part (see Figure 1) based on a multi-part diffusion framework. In Section 3.1, we outline the foundational vecset-based 3D latent diffusion model. Section 3.2 introduces our part-conditioning strategy using bounding box prompts and semantic point features, followed by the presentation of the complete $\mathcal{X}$-Part framework for synchronized part generation and its training scheme. Finally, we introduce the part editing pipeline in Section 3.3.

### 3.1 PRELIMINARY

Our method builds upon pre-trained vecset-based 3D shape generation models Zhang et al. (2024); Zhao et al. (2023; 2025); Li et al. (2025b), which typically consist of a 3D shape variational autoencoder (VAE) and a latent diffusion model.

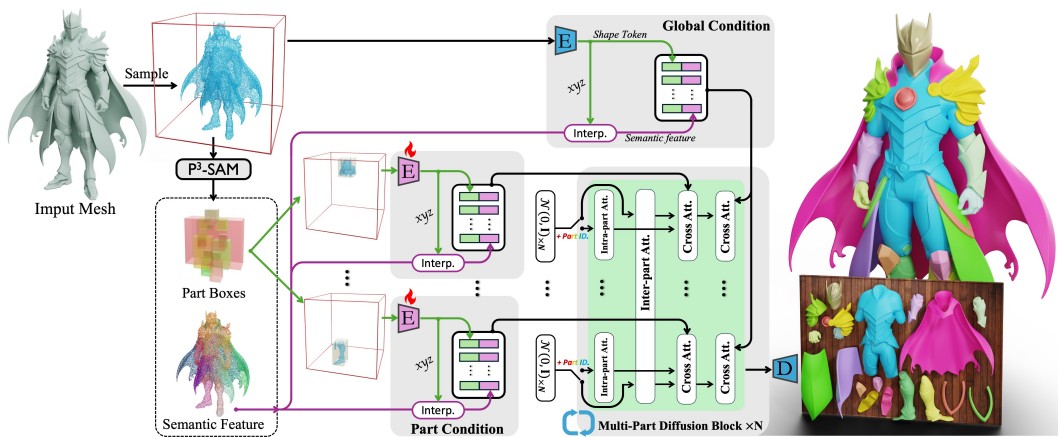

Figure 1: **Architecture of $\mathcal{X}$-Part**. Given the input point cloud, per-point feature and part bounding boxes are extracted from P³-SAM Ma et al. (2025). Global and part conditions are obtained by stacking geometry token with interpolated semantic features. They are injected to multi-part diffusion process to guide shape decomposition.

**Variational Autoencoder (VAE).** Following Zhao et al. (2025), given an input mesh, we first sample point cloud $\boldsymbol{X} \in \mathbb{R}^{N \times 7}$ including XYZ coordinates, surface normals, and a flag indicating if the point lies on a sharp edge. The encoder of the VAE consists of a cross-attention block and multiple self-attention layers. It maps the sampled point cloud into latent vectors:

$$\boldsymbol{Z} = \mathcal{E}(\boldsymbol{X}) = \text{SelfAttn}(\text{CrossAttn}(PE(\boldsymbol{X_0}), PE(\boldsymbol{X}))) \tag{1}$$

where $\boldsymbol{X_0} \in \mathbb{R}^{N_0 \times 7}$ denotes the point set obtained by applying farthest point sampling (FPS) to $\boldsymbol{X}$, and $\boldsymbol{Z} \in \mathbb{R}^{N_0 \times C}$ represents the $N_0$ latent tokens of the input shape. $PE$ represents position embedding for input point cloud. The decoder of the VAE similarly consists of several self-attention layers followed by a final cross-attention module, mapping a spatial coordinate query $q \in \mathbb{R}^3$ to its corresponding signed distance value (SDF). To enhance the capacity of VAE to represent part-level geometry, we further fine-tune the VAE on a dataset of part shapes.

**3D Diffusion Model.** To model the latent space of encoded objects, a flow-based diffusion model Lipman et al. (2022) is trained to generate latent tokens, which can subsequently be decoded into 3D geometries. Following Hunyuan-DiT Li et al. (2024) and TripoSG Li et al. (2025b), the core of our model is constructed using a series of Diffusion Transformer (DiT) blocks.

### 3.2 Multi-Parts Latent Diffusion

**Semantic-Aware Shape Conditioning.** To incorporate holistic shape information, we encode the input point cloud $\boldsymbol{X}$ using the VAE encoder, producing a global object condition $\boldsymbol{f}_o$ that encapsulates the complete geometric structure. To enable controllable part decomposition, we design a bounding box-driven conditioning module that extracts part-specific cues from the specified spatial regions, as illustrated in Figure 1. Specifically, we run P³-SAM Ma et al. (2025) to obtain part bounding boxes and per-point semantic features. Then, we sample points $\boldsymbol{X}_{\text{inbox}}$ within the given bounding box from the object point cloud. $\boldsymbol{X}_{\text{inbox}}$ is then encoded by a learnable encoder to form the part-level condition $\boldsymbol{f}_p$. To improve the robustness to bounding box perturbations during inference, we apply augmentations involving random translations and moderate scaling to the bounding boxes during training. To facilitate coherent shape decomposition, we enhance input conditions by concatenating shape tokens with semantic features. The enhanced object and part conditional features, $\boldsymbol{f}_o'$ and $\boldsymbol{f}_p'$, are defined as:

$$\boldsymbol{f}_o' = \text{Concat}(\boldsymbol{f}_o, Interp(\mathcal{E}_{sem}(\boldsymbol{X}), \boldsymbol{X})), \boldsymbol{f}_o = \mathcal{E}_o(\boldsymbol{X})$$
$$\boldsymbol{f}_p' = \text{Concat}(\boldsymbol{f}_p, Interp(\mathcal{E}_{sem}(\boldsymbol{X}), \boldsymbol{X}_{\text{inbox}})), \boldsymbol{f}_p = \mathcal{E}_p(\boldsymbol{X}_{\text{inbox}}) \tag{2}$$

where $\mathcal{E}_o$ denotes the raw shape VAE encoder which is frozen during training. $\mathcal{E}_p$ represents the learnable encoder in part condition extraction module. $\mathcal{E}_{sem}$ represents for the semantic encoder in

$P^3$-SAM. Note that to align with the shape tokens, the semantic feature is obtained by interpolated using the down-sampled XYZ positions from the shape encoder output, c.f. Figure 1. To enhance the robustness to the high-dimensional semantic feature, we apply random dropout for semantic feature. It is worth noting that when extracting the part-level condition, the bounding box of a specific part may contain points from adjacent parts. However, through the integration of point-wise semantic features and inter-part attention (described in Section 3.2), our model enables mutual exclusion of irrelevant points across different parts during the generation process.

**Multi-Part Diffusion.** We leverage multi-part diffusion to simultaneously generates latent tokens for all parts $O = Concatenate(\{z_i\}_1^K) \in \mathbb{R}^{nK \times C}$, where the object consists of $K$ parts and each part represented by $n$ latent tokens denoted as $z_i \in \mathbb{R}^{n \times C}$. Multi-part diffusion block repeats $N$ times and each block consists of one self-attention layer followed by two cross-attention layers (see Figure 1). At even blocks, self-attention is conducted within each part, providing intra-part awareness. At odd blocks, self-attention runs across all parts, exchanging inter-part information. This design aligns with Lin et al. (2025). Formally it reads

$$\mathbf{Attn}_{intra} = \mathbf{softmax}(\frac{\sigma_q(z_i)\sigma_k(z_i)^T}{\sqrt{d}})\sigma_v(z_i), \mathbf{Attn}_{inter} = \mathbf{softmax}(\frac{\sigma_q(z_i)\sigma_k(O)^T}{\sqrt{d}})\sigma_v(O) \quad (3)$$

where $\sigma_q$, $\sigma_k$, and $\sigma_v$ denote the query, key, and value projection layers, respectively, and $d$ represents the hidden dimension of the attention tokens. The global condition $f_o'$ and part conditions $f_p'$ are injected into the diffusion block by two cross-attention layers. We incorporate a learnable part embedding to further enhance the distinctiveness of each part. Specifically, we initialize a part embedding codebook $E \in \mathbb{R}^{l \times C}$ and assign a unique embedding to each part. A part embedding is repeated by $n$ and added to the part's token. To enable the decomposition of objects that contain more parts than the maximum limit for a single object in the training dataset, during training, we set $l$ to a much larger number, and randomly assign a unique embedding to each part.

**Training.** We train the model using the flow matching objective Lipman et al. (2022). During the forward process, Gaussian noise $\varepsilon \sim \mathcal{N}(0, \mathbf{I})$ is added to the data $z_0$ according to a noise level $t$, resulting in $z_t = tz_0 + (1 - t)\varepsilon$. The model is trained to predict the velocity field $v = \varepsilon - z_0$ that moves $z_t$ back toward $z_0$, conditioned on both the global condition $f_o'$ and the part condition $f_p'$.

$$\mathcal{L} = \mathbb{E}_{z,t,\varepsilon} \left[ \left\| (\varepsilon - z_0) - v_\theta(z_t, t, f_o', f_p') \right\|^2 \right] \quad (4)$$

where $v_\theta$ denotes the denoising neural network. Given that the geometric complexity of an individual part is substantially lower than that of a complete object, we assign a reduced number of tokens to each part during both the VAE fine-tuning process and $\mathcal{X}$-Part training process.

### 3.3 Part Editing

We further design a interactive part editing pipeline based on X-Part. Following Repaint Lugmayr et al. (2022), we adopt a training-free method to achieve two kinds of editing: part split and part adjust. The split operation refers to splitting the bounding box and generating several parts accordingly. The adjust operation means adjusting a certain bounding box so that the part and adjacent parts would be re-generated accordingly. Specifically, for parts indicated by the bounding box, their latent tokens are resampled and denoised while keeping tokens of other parts unchanged.

## 4 Experiments

**Evaluation Metrics.** We evaluate our method on 200 samples from the ObjaversePart-Tiny dataset, each comprising rendered images and corresponding ground-truth part geometries. To assess geometric quality, we employ Chamfer Distance (CD) and F-Score. The F-Score is computed at two different thresholds $[0.1, 0.05]$ to capture both coarse-level and fine-level geometric alignment. Prior to metric computation, each object is normalized to the range $[-1, 1]$. To ensure pose-agnostic evaluation, we rotate each object by $[0, 90, 180, 270]$ degrees and report the best score among these orientations as the final metric.

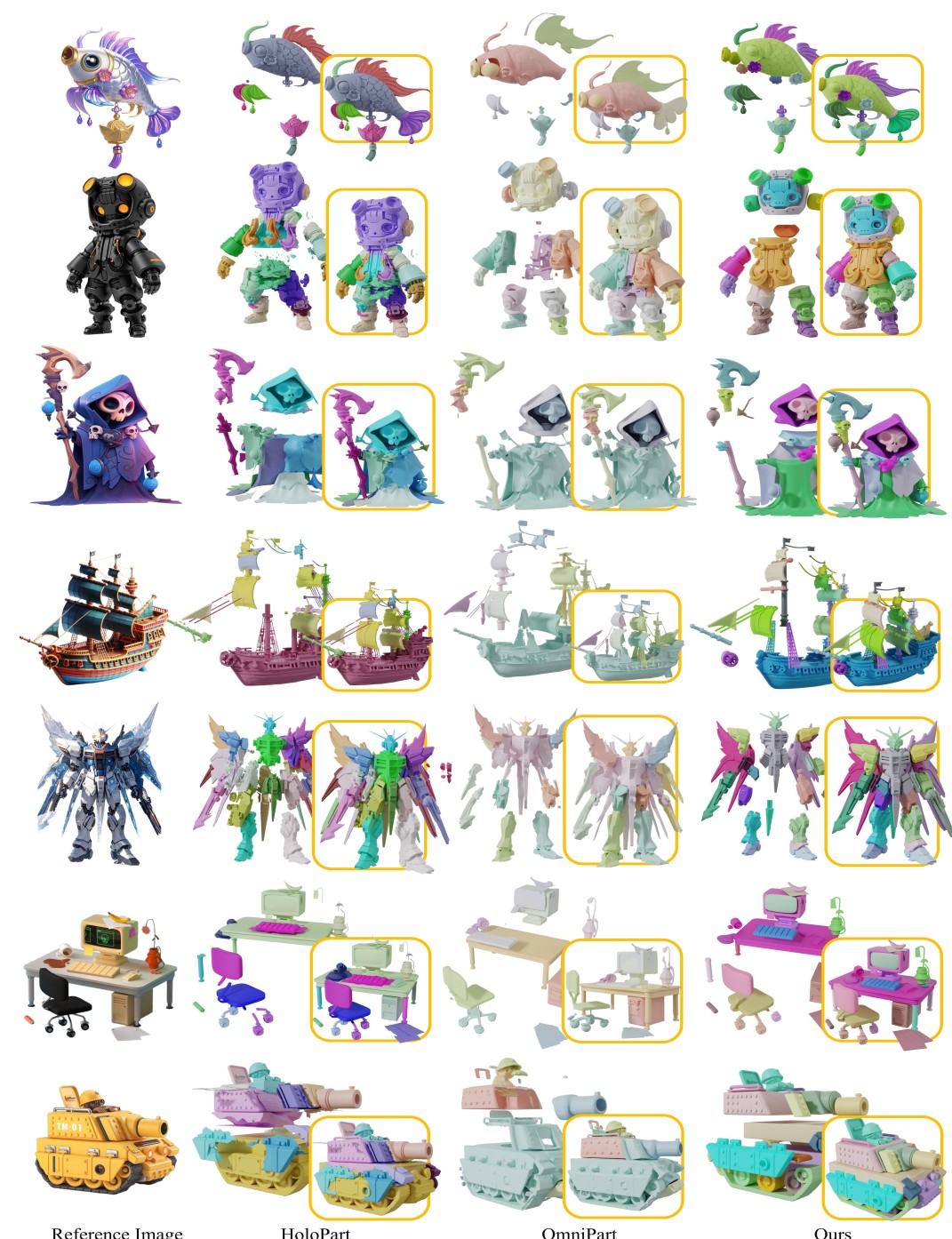

Figure 2: **Qualitative shape decomposition results.** Note that the input shapes for our method and HoloPart are ground-truth watertight point clouds, while OmniPart leverages shapes produced by Trellis Xiang et al. (2025).

**3D Shape Decomposition results.** This experiment aims to evaluate and compare the geometric decomposition capabilities of different methods, validating that our approach achieves a deeper structural understanding and decomposition of objects while generating higher-quality part geometries. Our method takes a ground-truth watertight surface as input and automatically generates decomposed parts; We compute metrics between the generated parts and the ground-truth

parts. We first compare against segmentation-based methods such as Sampart3D Yang et al. (2024) and PartField Liu et al. (2025), which also take the same watertight mesh as input. The segmented results are directly compared to the ground truth parts. In addition, we include generative methods such as HoloPart Yang et al. (2025a) and OmniPart Yang et al. (2025b). HoloPart also uses the ground-truth watertight point cloud as input. Although OmniPart does not directly take a 3D shape as input, it first generates a coarse geometry and then performs part decomposition. To eliminate the influence of segmentation quality, we replace the Sampart3D segmentation used in HoloPart with $P^3$-SAM Ma et al. (2025), and provide OmniPart with 2D part masks rendered from the ground-truth parts. As shown in Table 1, segmentation-based methods can decompose part points on the input watertight surface but fail to produce complete part geometries. Our method outperforms all baselines in decomposition quality, even when OmniPart is supplied with ground-truth 2D masks. Furthermore, as illustrated in Figure 2, our approach significantly surpasses other methods in the geometric quality of the generated parts.

| Method | CD↓ | Fscore-0.1↑ | Fscore-0.05↑ |
|---|---|---|---|
| SAMPart3D | 0.15 | 0.73 | 0.63 |
| PartField | 0.17 | 0.68 | 0.57 |
| HoloPart | 0.26 | 0.59 | 0.43 |
| OmniPart | 0.23 | 0.63 | 0.46 |
| Ours | **0.11** | **0.80** | **0.71** |

Table 1: Part decomposition results.

**Image-to-3D Part Generation.** Leveraging existing image-to-3D generative models, we extend our method to the task of image-to-3D part generation. Specifically, given a reference image, we first generate a watertight mesh using an off-the-shelf image-to-3D model Zhang et al. (2024); Lai et al. (2025); Li et al. (2025b), which is then fed into our pipeline for decomposition into parts. Similar to the previous experiment, we compare our approach not only against HoloPart and OmniPart, but also against methods that directly generate parts from images, such as PartPacker Tang

| Method | CD↓ | Fscore-0.1↑ | Fscore-0.05↑ |
|---|---|---|---|
| Part123 | 0.42 | 0.36 | 0.20 |
| HoloPart | 0.09 | 0.88 | 0.73 |
| PartCrafter | 0.20 | 0.66 | 0.45 |
| PartPacker | 0.11 | 0.85 | 0.65 |
| OmniPart | **0.08** | 0.91 | 0.77 |
| Ours | **0.08** | **0.92** | **0.78** |

Table 2: Holistic shape generation results.

et al. (2025), PartCrafterLin et al. (2025), and Part123Liu et al. (2024). The input to OmniPart remains consistent with the setup above, while both HoloPart and our method use the same generated mesh as input. Since different methods may produce divergent part structures, making it difficult to establish accurate correspondences with ground-truth parts. We compare only the overall object geometry composed of all generated parts. As shown in Table 2, our method produces final objects with higher geometric quality and better alignment to the ground truth. Figure 2 visually demonstrates the structural plausibility and high quality of our results. Moreover, our decomposition is more refined, often generating a larger number of semantically reasonable parts.

**Part Editing.** In Figure 4(a), we demonstrate the two types of part editing methods as described in Section 3.3, which demonstrates the controllability of our proposed method.

**Part-Aware UV Un-wrapping.** UV unwrapping is an essential step in 3D content creation pipelines. Fig. 4 compares the UV maps generated by unwrapping a holistic mesh and part-decomposed meshes respectly. Part-decomposed mesh are processed by unwrapping each of the part separatedly. Decomposing shapes into part greatly simplify Un-wrapping process and makeing UV maps more compact and semantically meannful.

**Ablation Study** As shown in Table 3, we conduct a series of ablation studies to validate the effectiveness of each component in our proposed framework, all of which contribute to improved model performance. We analyze the roles of individual components in detail. The intra-part and inter-part attention mechanism enhances the representation of part-level latents while maintaining a global contextual view across all parts. The part embedding module introduces distinctiveness among the latent representations of different parts. The object-level condition provides priors about the overall geometry of the shape. Meanwhile, the part-level condition offers detailed information indicating coarse part location and scale. Additionally, the semantic point feature supplies semantic cues that facilitate structurally coherent shape decomposition. We further provide visualizations of representative results in Figure 5 to illustrate the impact of each component.

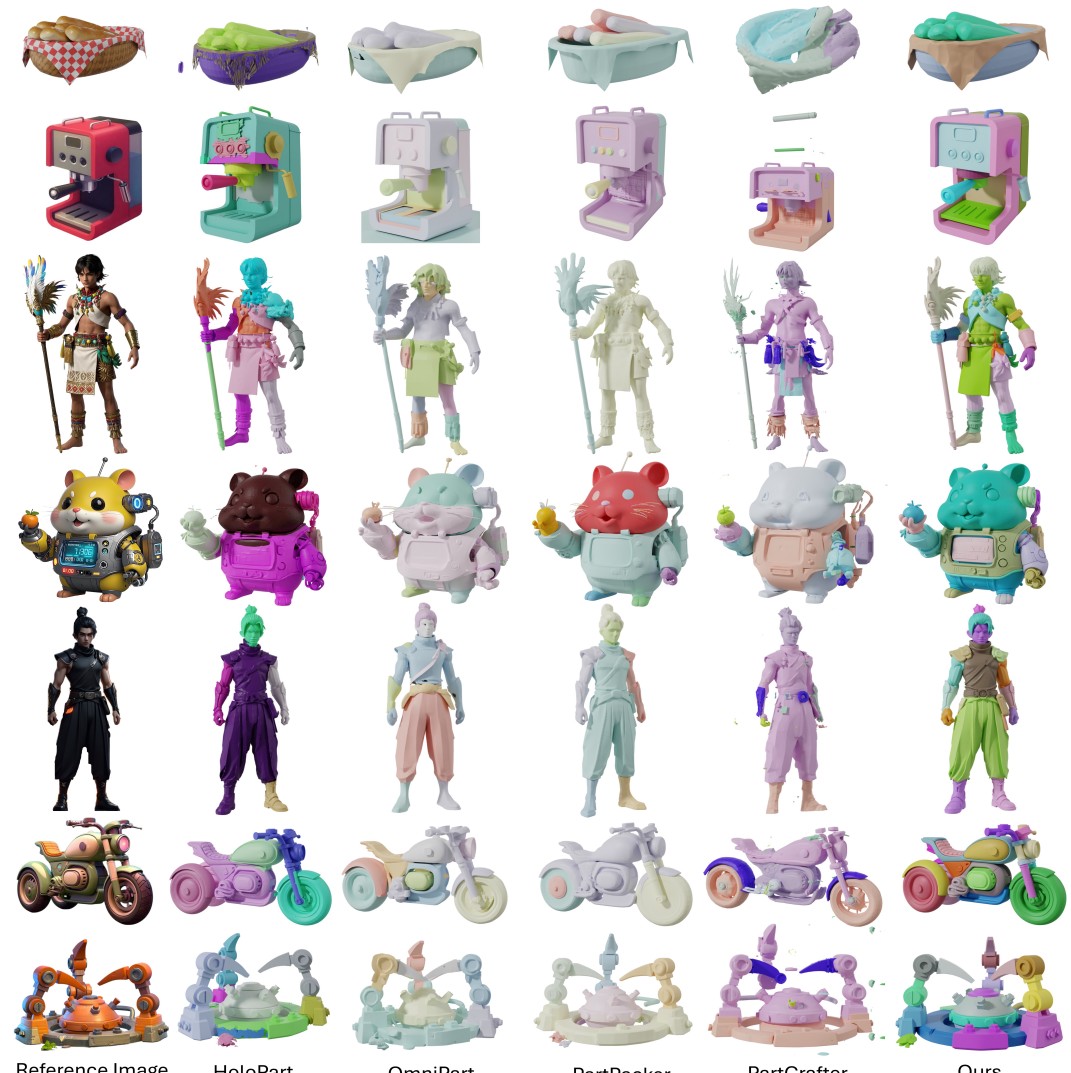

Figure 3: **Qualitative shape decomposition results.** Note that the input shapes for HoloPart and Ours are obtained from Hunyuan3D-2.5 Lai et al. (2025), while OmniPart leverages shapes produced by Trellis. PartCrafter and PartPacker do not rely on shapes.

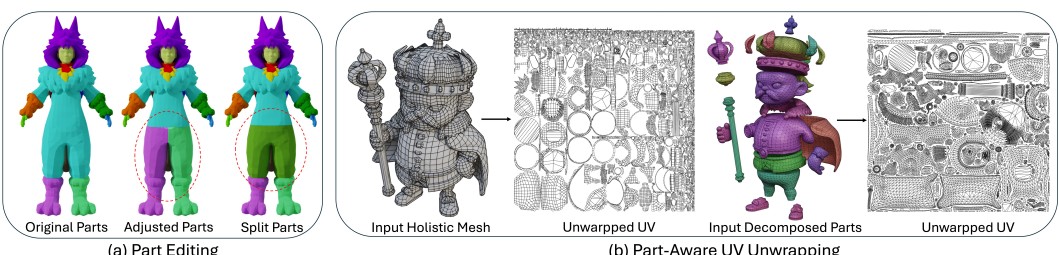

Figure 4: Demonstration of two representative applications of our method. Subfigure (a) shows the results of bounding box-controlled part generation, while subfigure (b) illustrates improved UV unwrapping performance achieved through part-based decomposition.

| Method | Part-level | | | Overall-level | | |
|---|---|---|---|---|---|---|
| | CD ↓ | F1-0.1 ↑ | F1-0.05 ↑ | CD ↓ | F1-0.1 ↑ | F1-0.05 ↑ |
| W/o part embedding | 0.13 | 0.78 | 0.68 | 0.04 | 0.97 | 0.92 |
| W/o object-cond | 0.12 | 0.79 | 0.70 | 0.03 | 0.97 | 0.93 |
| W/o part-cond | 0.27 | 0.57 | 0.47 | 0.03 | **0.98** | 0.95 |
| W/o semantic-feat | 0.12 | 0.78 | 0.69 | 0.04 | 0.97 | 0.92 |
| W/o inter-part self-attn | 0.12 | 0.79 | 0.70 | 0.03 | 0.97 | 0.94 |
| Ours | **0.11** | **0.80** | **0.71** | **0.02** | **0.98** | **0.96** |

Table 3: Based on the ground-truth bounding boxes, we compute part-level and object-level metrics for different modules on the ObjaversePart-Tiny dataset.

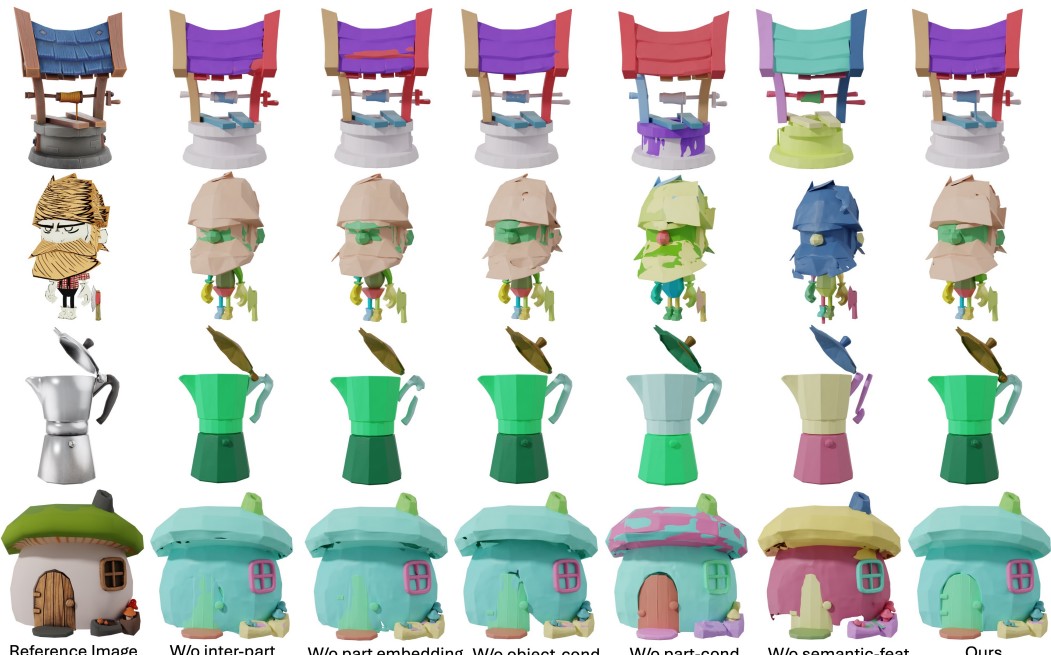

Figure 5: Part generation results under different module ablation settings.

## 5 CONCLUSION AND LIMITATION

**Conclusion** We introduce $\mathcal{X}$-Part, a purely geometry-based part generation framework that takes bounding boxes as input to decompose complete 3D objects into structured parts. Compared to existing approaches, our method better preserves geometric quality and fidelity in the generated parts, while also offering easier integration into 3D content creation pipelines, thereby significantly reducing the complexity of downstream tasks. Additionally, our method allows users to alter part decomposition strategies by adjusting bounding boxes, thereby enabling more intuitive control and flexible editing. To enhance the model's structural understanding, we incorporate semantic point features that provide high-level shape semantics. Our approach supports the generation of up to 50 distinct parts, which sufficiently covers most practical application scenarios.

**Limitation** Our method currently relies on geometric cues for decomposition and lacks guidance from physical principles, which may limit its ability to meet certain application-specific decomposition requirements. Additionally, since the latent codes of all parts are processed simultaneously through the diffusion model, inference time increases with the number of parts, posing a challenge for real-time usage when handling high-part-count objects.

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

## A  APPENDIX

### A.1  THE USE OF LARGE LANGUAGE MODELS (LLMS)

All technical contributions, including the methodology, equations, and results, are solely the work of the authors.

### A.2  IMPLEMENTATION DETAILS

**Network Architecture** The DiT module consists of 21 DiT blocks, where skip connections are implemented by concatenating latent features along the channel dimension. During training, the number of tokens per part is set to 512, consistent with the VAE fine-tuning configuration. The self-attention layers at odd indices are configured to perform inter-part attention, thereby enhancing awareness of other parts. For the cross-attention modules, both the object condition and the part condition are represented with 2,048 tokens, providing detailed guidance for the generation process. The part embedding codebook contains 50 entries, and a unique embedding is randomly assigned to each part latent during both training and inference. In addition, we employ a Mixture-of-Experts (MoE) model for the linear output layers of the first six network blocks to efficiently enhance the learning capacity in the latent space.

**Training** Our model is initialized from a pre-trained object generator, with its self-attention parameters loaded as the starting point. We use the Adam optimizer with a learning rate of $1e-4$ and apply gradient clipping with a maximum norm of $1.0$ to enhance training stability. The model was trained for approximately four days on 128 H20 GPUs. To further improve robustness, we randomly drop semantic features with a probability of $0.3$, and independently apply a $0.1$ dropout probability to the object condition, the part condition, or both during training. Additionally, we apply data augmentation to the bounding boxes by introducing random translations sampled from a uniform distribution $\mathcal{U}(-0.05, 0.05)$ and scaling factors sampled from the interval $[0.9, 1.1]$.

**Dataset Curation** We use the part dataset introduced in $P^3$-SAM Ma et al. (2025), which contains nearly 2.3 million objects with ground truth part segmentation. To create training pairs, each part of an object, as well as the object itself, is remeshed into a watertight mesh. A dataset of this scale significantly enhanced the generalizability of our diffusion-based shape decomposition method.

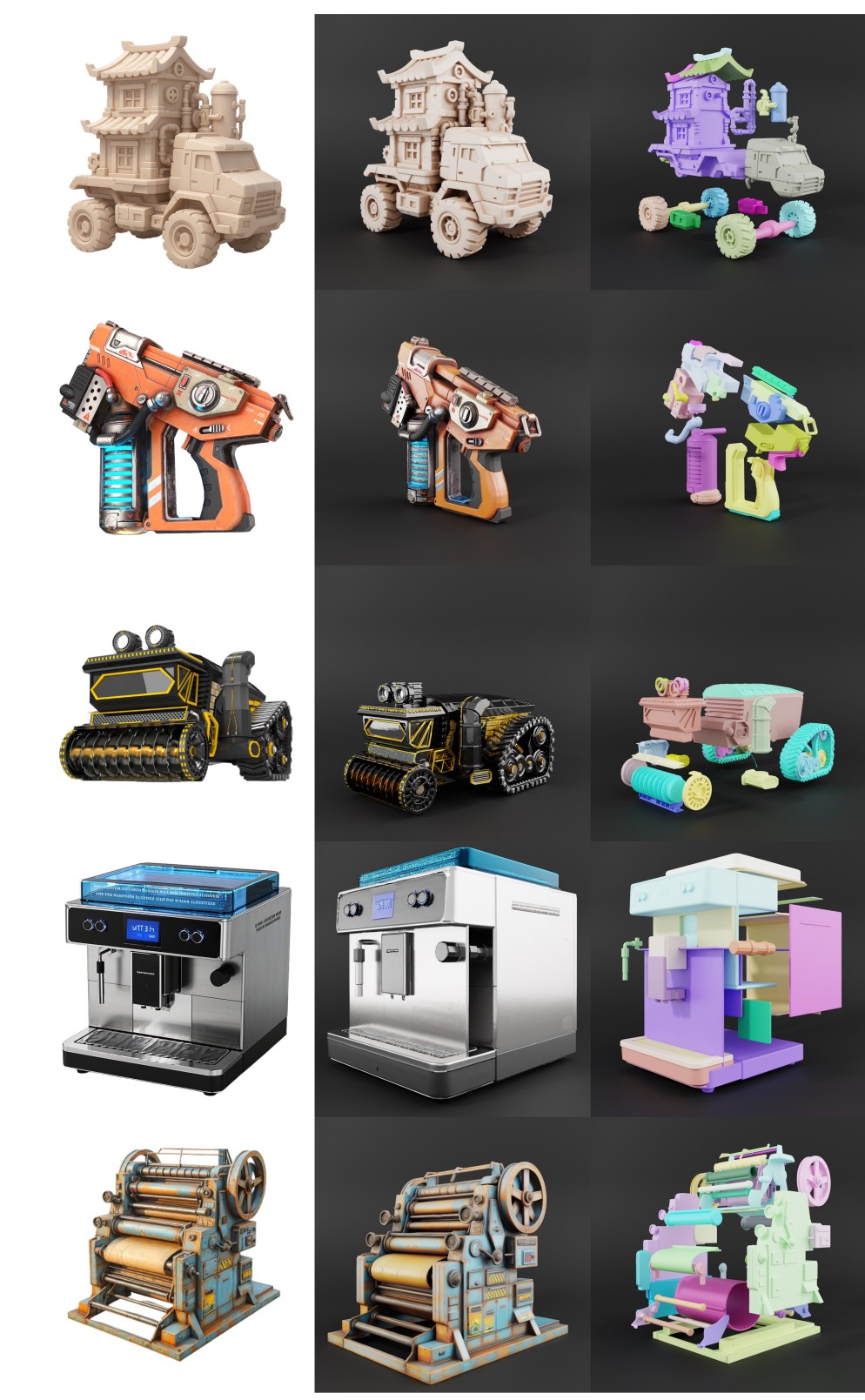

Figure 6: More results. The left column shows the input images, the middle column displays the object meshes generated by Hunyuan3D-2.5 Lai et al. (2025), and the right column presents the decomposition results obtained by our $\mathcal{X}$-Part framework.

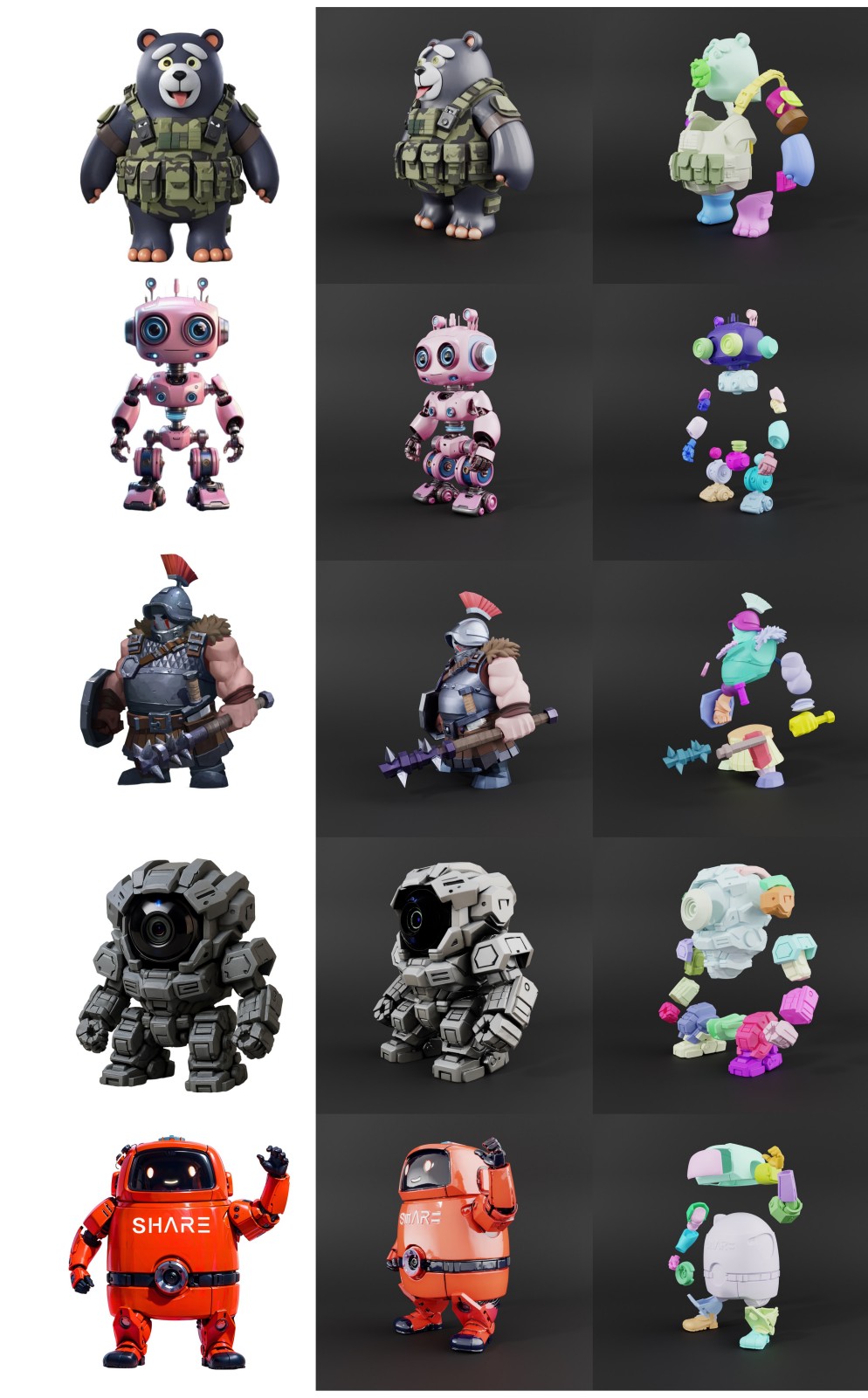

Figure 7: More results. The left column shows the input images, the middle column displays the object meshes generated by Hunyuan3D-2.5 Lai et al. (2025), and the right column presents the decomposition results obtained by our $\mathcal{X}$-Part framework.

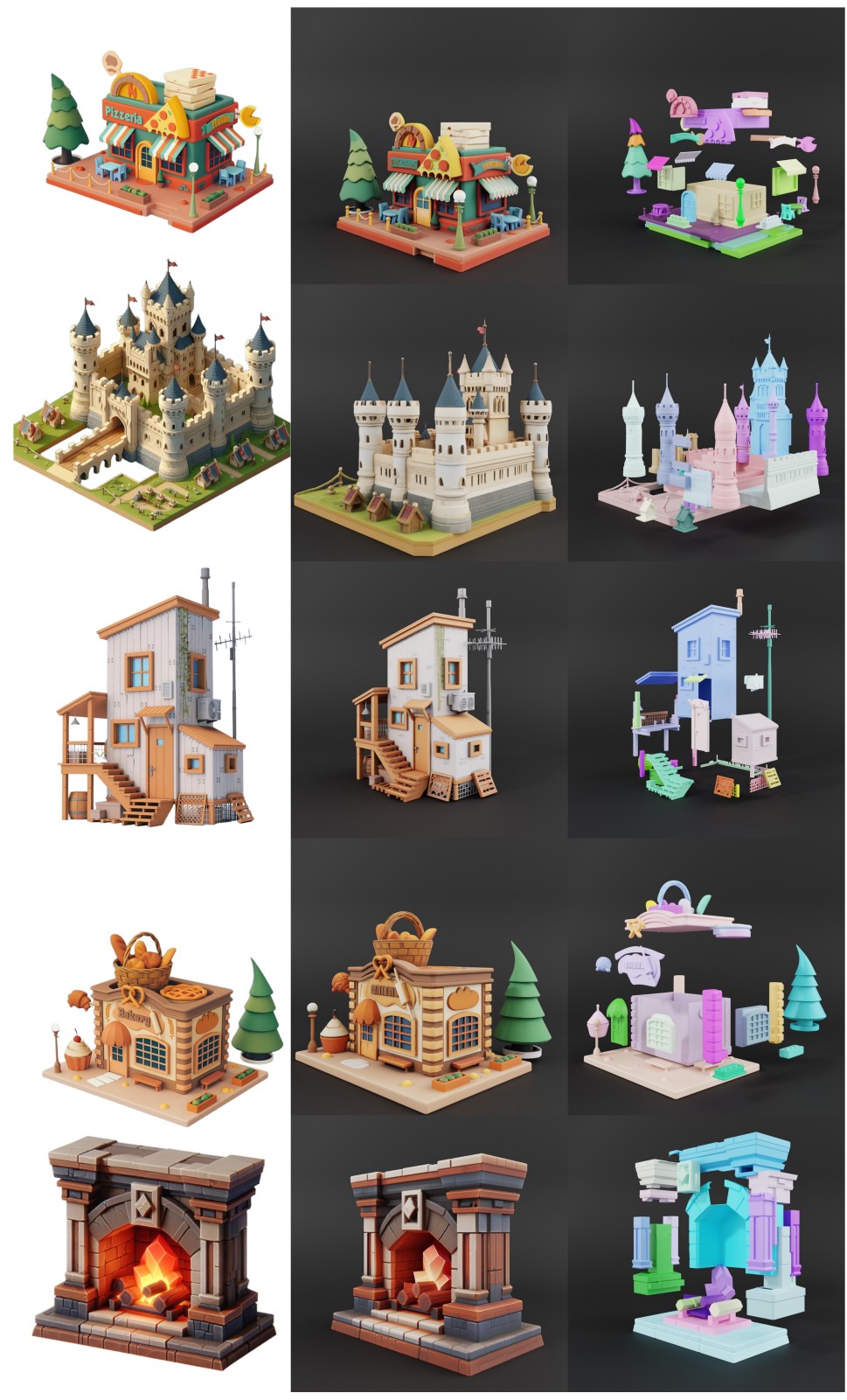

Figure 8: More results. The left column shows the input images, the middle column displays the object meshes generated by Hunyuan3D-2.5 Lai et al. (2025), and the right column presents the decomposition results obtained by our $\mathcal{X}$-Part framework.

