# OpenReview forum: "X-Part: High-fidelity and Structure-coherent Shape Decomposition"
_ICLR.cc/2026/Conference — ICLR 2026 Conference Withdrawn Submission_

### Official Review · Reviewer_4YZ4 · 2025-10-26

**Soundness:** 2
**Presentation:** 3
**Contribution:** 2
**Rating:** 2
**Confidence:** 4

**Summary:**

This paper proposes X-Part, a diffusion-based framework for decomposing holistic 3D shapes into semantically meaningful and structurally coherent parts. The method builds directly on P3-SAM, using it to extract bounding boxes and per-point semantic features as conditions for a multi-part latent diffusion model. Experiments on ObjaversePart-Tiny demonstrate improved Chamfer Distance and F-scores over segmentation and part-level generative baselines.

**Strengths:**

- The paper is mostly well-written and easy to read.
- The proposed pipeline is clear and leverage existing foundation models like P3-SAM. It integrates P3-SAM’s strong segmentation capability with diffusion–based part generation framework. However, this might harm this paper's own novelty.
- Quantitative improvement on benchmarks. The reported results show gains over prior baselines, suggesting that diffusion-based refinement may help produce smoother and more complete part geometries.

**Weaknesses:**

- **Incremental novelty relative to P3-SAM.**
The framework heavily depends on P3-SAM for both segmentation and semantic features. Since P3-SAM already outputs per-vertex part labels and embeddings, the proposed diffusion stage largely performs post-refinement—transforming segmented vertices into individual mesh parts. Conceptually, this appears to be a modest extension rather than a fundamentally new decomposition method.

- **Lack of physical plausibility or structural reasoning.**
The generated parts are evaluated only with geometric reconstruction metrics (Chamfer, F-score). There is no discussion of physical plausibility, e.g., whether adjacent parts intersect or leave gaps. Since the paper emphasizes “structurally coherent” decomposition, it should include evaluations such as part-part collision checks, inter-part penetration ratios, or joint-surface alignment.

- **Overstated claims of controllability.**
The paper listed “interactive editing” as one of its core contributions but the operations are simple bounding-box resampling steps without semantic understanding or global consistency guarantees. These do not yet amount to a robust editing framework.


- **(Minor) Citation.** citet and citep commands are not correctly used. The bib entry for paper *Repaint: Inpainting using denoising diffusion probabilistic models* emerges twice in the reference section. Please double check the citation format.

**Questions:**

- The method is trained on the large-scale 3D asset dataset released in P3-SAM and evaluated on Objaverse-Tiny. However, from the P3-SAM, it seems ObjaverseTiny is included in P3-SAM. Is this the case? If so, the experimental results would be doubtful.

- Efficiency and complexity analysis, including inference time, GPU requirements, and scaling trends with number of parts would be useful.

---

### Official Review · Reviewer_nU55 · 2025-10-27

**Soundness:** 3
**Presentation:** 4
**Contribution:** 3
**Rating:** 8
**Confidence:** 5

**Summary:**

This paper introduces X-Part, a diffusion-based framework for decomposing a holistic 3D object into semantically meaningful and structurally coherent parts with high geometric fidelity. The method utilizes bounding boxes as prompts for part generation and injects point-wise semantic features to guide the decomposition within a synchronized multi-part diffusion process. X-Part also includes a pipeline for interactive part editing.
Overall, this is a strong paper proposing a novel and effective method for a crucial task in 3D content creation. The focus on high fidelity, structural coherence, and controllability addresses key limitations of existing approaches.

**Strengths:**

- X-Part significantly advances the state-of-the-art in both part shape quality and semantic correctness. The extensive quantitative and qualitative results (Tables 1, 2, Figures 2, 3) clearly support this claim, especially its superior performance in part decomposition metrics (Table 1).

- The use of bounding boxes as prompts and the inclusion of an interactive part editing pipeline (part split and part adjust) provide essential user control, which is highly valuable for production-ready 3D asset creation.

- The demonstration of Part-Aware UV Unwrapping highlights the practical utility of the structured part decomposition for downstream tasks, making the generated assets more production-ready.

**Weaknesses:**

- The method heavily relies on $P^3$-SAM for initial part segmentations, bounding boxes, and semantic features. Any potential inaccuracies or failures in $P^3$-SAM could impact the quality of the X-Part decomposition, despite the use of coarse bounding boxes and robust semantic features. How P3-san is robust to support the fine-grained part parsing?

- The authors note that inference time increases with the number of parts because the latent codes for all parts are processed simultaneously in the diffusion model. This poses a challenge for real-time usage or objects with a very high part count. The maximum supported number of parts is 50, which may be limiting for extremely complex assemblies.  Is this a hard limit imposed by the training setup (e.g., codebook size) or the scalability issue, and could the architecture theoretically handle a higher number of parts by increasing the codebook size? I am curious about the results on the keyboard, could the model can parse any key part from the keyboard?

- The current framework is purely geometry-based and lacks guidance from physical principles. This may limit its ability to meet specific, application-driven decomposition requirements (e.g., separating parts that move or interact physically).

- You mention assigning a reduced number of tokens to each part compared to the complete object. What is the quantitative basis for setting the number of tokens per part to 512? What is the impact on geometric fidelity if this number is further reduced, and how does the total memory consumption compare to a monolithic object generation model?


- The method avoids overfitting to fine-grained segmentation by using bounding boxes. Have you conducted an ablation that explicitly measures the method's performance when the $P^3$-SAM segmentation is deliberately noisy or less accurate, versus when only bounding boxes are slightly perturbed (as done during training)?

- The paper claim state-of-the-art performance in semantic correctness. While F-Score and Chamfer Distance (CD) measure geometric quality, they don't directly quantify semantic correctness. Was a semantic segmentation metric (like mIoU or part accuracy) used in an evaluation that is not shown in the provided tables? If not, please clarify how the claim of improved semantic correctness is quantitatively justified. A user study is very important for the quality and interactive editing tools.

- The editing pipeline supports "part split" and "part adjust" by resampling and denoising latent tokens for the selected bounding box region. Can the system also support merging parts or modifying the part geometry without changing the bounding box, which would be useful for adding fine details?

- The impressive work BANG also support the part parsing for any object, could there be a comparison with the SOTA work? BTW, for the part level 3D shape generation, there are lots of research work on the structure formulation and part detailed geometry, such as the StructureNet, Grass, DSG-Net, and so on. I think there should be an overview and discussion in the related work section.

- Since the paper mentioned that Part-Aware UVunwarping, what about the part-level texture generation for the part parsing. Could you display some textured part after the part parsing. (All exampels are using the different color to textured the parts.)

**Questions:**

See weaknesses

---

### Official Review · Reviewer_2ogY · 2025-10-28

**Soundness:** 3
**Presentation:** 3
**Contribution:** 2
**Rating:** 4
**Confidence:** 2

**Summary:**

The paper introduces X-Part, a shape decomposition pipeline built on top of P3-SAM. X-Part takes in an input mesh and generates each part as an SDF field using latent diffusion with a customized diffusion network that looks at both the part-level and global-level shape information. The network achieves SOTA performance og shapr decomposition and enables tasks such as smarter UV unwrapping.

**Strengths:**

1. The paper's result is impressive both in the numbers shown and the qualitative results presented.
2. The paper's evaluation is extensive, comparing with many SOTA methods on different metrics.

**Weaknesses:**

1. Comparison against baselines with equally sized data would be great to have, since it's unclear right now whether the result improvement comes from larger-scale training data or the network.
2. The method is unclear. Specifically, what is decoded at the end after the diffusion? Is it independent decoding of each part or the shape as a whole? How are the parts assembled?
3. The ablation study raises questions about the effectiveness of many design choices the paper makes. Seems that many components only led to marginal improvements across all metrics. This makes the technical contribution less convincing. Further, since the metrics reported in the ablation are already higher than the SOTA performance as reported in Tab. 1, it makes me question further whether the improvement is from larger-scale training.

**Questions:**

1. Does the method work on slightly noisy inputs, such as those from image/text->3D models? The geometry showcased in Fig. 2 is very clean, and I don't think it represents the average shape generation quality from these models.
2. Since there is a way to control the number of output parts, do the parts exhibit a hierarchical structure when the part number increases?

---

### Official Review · Reviewer_afpD · 2025-10-29

**Soundness:** 3
**Presentation:** 2
**Contribution:** 2
**Rating:** 6
**Confidence:** 4

**Summary:**

TLDR: combining P3-SAM and PartsCrafter with some technical adaptations.

This paper presents an approach to use segmented point clouds as conditions to generate 3D shape parts. It first uses P3-SAM to segment the point cloud, obtaining the bounding boxes of the parts. Then it uses the architecture from PartsCrafter to generate the parts within the bounding boxes. In addition to the part points as local conditions, it also uses a global shape condition to guide the part generation.

**Strengths:**

- Good results. Thanks to the bounding box guidance, instead of point cloud segments that could have broken boundaries, the parts are generated with fewer artifacts.
- Sound technical approach. The general design of the method is sound.

**Weaknesses:**

- Core claim of interactive part editing is very limited (shown in a sub-figure with a single example)
- Novelty is mediocre (P3-SAM + PartsCrafter), essentially just PartsCrafter with part-level conditions. This paper does not bring sufficient new insights, findings, or thoughts to the field.
- Insufficient analysis of the results. For example, why do HoloPart and OmniPart produce worse quality?

**Questions:**

It is not clearly illustrated how interactive part editing works and how well it will work in practice.
As a core claim of contribution, can you provide a more comprehensive demonstration and analysis?

---

### Note · Authors · 2025-11-13

I have read and agree with the venue's withdrawal policy on behalf of myself and my co-authors.